

# Appraisal of unimodal cues during agonistic interactions in *Maylandia zebra*

Laura Chabrolles[1], Imen Ben Ammar[1], Marie S.A. Fernandez[1,2], Nicolas Boyer[1], Joël Attia[1], Paulo J. Fonseca[3], M. Clara P. Amorim[4] and Marilyn Beauchaud[1]

[1] Equipe Neuro-Ethologie Sensorielle, ENES/Neuro-PSI CNRS UMR 9197, Université de Lyon/Saint-Etienne, Saint-Etienne, France
[2] INRIA, Beagle, Université de Lyon, Villeurbanne, France
[3] Departamento de Biologia Animal and cE3c—Centre for Ecology, Evolution and Environmental Changes, Faculdade de Ciências, Universidade de Lisboa, Lisbon, Portugal
[4] MARE—Marine and Environmental Sciences Centre, ISPA, Instituto Universitário, Lisbon, Portugal

## ABSTRACT

Communication is essential during social interactions including animal conflicts and it is often a complex process involving multiple sensory channels or modalities. To better understand how different modalities interact during communication, it is fundamental to study the behavioural responses to both the composite multimodal signal and each unimodal component with adequate experimental protocols. Here we test how an African cichlid, which communicates with multiple senses, responds to different sensory stimuli in a social relevant scenario. We tested *Maylandia zebra* males with isolated chemical (urine or holding water coming both from dominant males), visual (real opponent or video playback) and acoustic (agonistic sounds) cues during agonistic interactions. We showed that (1) these fish relied mostly on the visual modality, showing increased aggressiveness in response to the sight of a real contestant but no responses to urine or agonistic sounds presented separately, (2) video playback in our study did not appear appropriate to test the visual modality and needs more technical prospecting, (3) holding water provoked territorial behaviours and seems to be promising for the investigation into the role of the chemical channel in this species. Our findings suggest that unimodal signals are non-redundant but how different sensory modalities interplay during communication remains largely unknown in fish.

# INTRODUCTION

Research on animal conflict is critical to our understanding of social interactions. A common cause of conflict is the dispute for territories which can be key in acquiring food or mates (*Hurd & Enquist, 2001*). Agonistic interactions during territorial contests can provoke serious injuries, but contestants can also avoid costly fights by signalling their status and performing threat displays to elicit the opponent's withdrawal (*Hurd & Enquist, 2001*; *Van Staaden, Searcy & Hanlon, 2011*). Communication is thus fundamental in resolution of animal conflicts.

Corresponding author
Laura Chabrolles,
l.chabrolles@univ-st-etienne.fr

Animals communicate through multiple senses which allow them to detect and integrate information during social interactions. To better understand animal communication, we must consider all the different sensory modalities involved. It is well known that communication is a multiple sensory system (*Darwin, 1998*), but recently there is a growing interest in understanding how different sensory channels are integrated to produce adaptive behaviour (*Partan & Marler, 2005*; *Higham & Hebets, 2013*). A first step towards the understanding of how different modalities interact during communication, is to investigate the behavioural response in a 'cue-isolation' experiment (*Partan & Marler, 1999*; *Smith & Evans, 2013*). Several communication channels are known to be widely used by fish. Many authors have proposed that visual (*Rosenthal & Ryan, 2000*; *Dijkstra et al., 2010*; *Chen & Fernald, 2011*; *Mellor et al., 2012*), acoustic (*Amorim et al., 2008*; *Bertucci et al., 2010*; *Maruska, Ung & Fernald, 2012*; *Longrie et al., 2013*) and chemical cues (*Plenderleith et al., 2005*; *Martinovic-Weigelt et al., 2012*; *Keller-Costa, Canário & Hubbard, 2015*) are highly relevant for intraspecific communication. Additional studies revealed the importance of the lateral line system during social encounters (*Butler & Maruska, 2015*). However, much remains unknown regarding the role of each sensory modality in eliciting behavioural decisions, particularly their relative hierarchy during multi-modal sensory integration. For example, acoustic and chemical cues are rarely analysed alone and typically need a visual trigger to elicit a behavioural response (*Estramil et al., 2014*; *Bayani, Taborsky & Frommen, 2017*). The aim of this work was to test with a 'cue-isolation' experiment the prioritization of sensory modalities by investigating how fish respond to isolated chemical, visual and acoustic signals in an agonistic context. It further aimed to develop a bioassay to test the role of each sensory channel on behavioural decisions by assessing the response to different stimulus types, e.g., video playback vs. live male stimulus.

African cichlids use multimodal signals to communicate during social interactions and became important models to evaluate the relative importance of different sensory channels (*Escobar-Camacho & Carleton, 2015*). *Maylandia zebra* is a lek-breeding cichlid living in the sediment-free rocky coast of Lake Malawi. Males of *M. zebra* defend territories to which they try to attract females by means of conspicuous displays that involve multiples senses (*Plenderleith et al., 2005*; *Miguel Simões et al., 2008*). These territories are not only used for mating but also for feeding, and males defend them from other males for prolonged periods (*Holzberg, 1978*). In this study, we exposed dominant males to the following stimuli: chemical (urine or holding water from dominant males; teleost Ringer solution was used as a control), visual (a real opponent in a jar or video playback), and acoustic (the playback of an agonistic sound). Based on previous studies (*Bertucci et al., 2010*; *Maruska & Fernald, 2012*), we predicted that vision would trigger direct agonistic behaviours and acoustic or chemical stimuli alone would rather induce indirect territorial behaviours such as increased activity and nest building.

## MATERIALS & METHODS

### Subjects and housing conditions

Seventy *Maylandia zebra* were purchased from Oxyfish (Verlinghem, France) and stored in holding tanks (120 cm long, 60 cm wide and 50 m height) at ENES laboratory (University of

Saint-Etienne, France). Each tank contained 10 males that were fed daily with commercial cichlid food (JBL NovoRift sticks and Tetra flakes). All tanks were equipped with an external filter (Rena Filstar xP3; Rena France, Annecy, France), aeration and PVC tubes as shelters. The water temperature was maintained at 25 ± 1 °C with a pH of 8.0 and a 12:12 h light: dark cycle was settled. Each fish was identified by a transponder (PSK Transponder; Dorset Identification B.V., Dorset, Netherlands).

## General experimental design

The experimental aquarium (60 cm × 30 cm × 30 cm) was placed on a vibration-insulated shelf (with a layer of wood-fibre) in a sound-proofed chamber (PRIMO Silence-Box, Tip Top Wood, Saint-Etienne, France) to limit background noise. The back of the aquaria was covered on the inside with bubble wrap to break sound reflections on the tank's walls.

Each experimental tank had a filter containing active carbon, aeration, an internal heater, sand substrate, and a terracotta pot usable as a shelter by the fish, all located on the right side of the tank. Because the tested fish stayed in the experimental set up for five days, each aquarium was accompanied by a social group placed in an adjacent tank, to avoid the sense of isolation. These social groups were composed of one male and one female physically separated to avoid any injuries. An underwater loudspeaker developed by *Fonseca & Alves (2012)* was placed in the aquarium of the tested fish and used for playing back sound stimuli. A hydrophone (Aquarian Audio Products H2a-XLR; AFAB Enterprises, Anacortes, WA, USA, sensitivity: −180 dB re 1 V μPa-1, flat frequency response ± 4 dB 20 Hz–4.5 kHz) located in the middle of the tank, monitored sound playback and registered possible sounds produced by the tested fish.

The hydrophone was connected to a preamplifier (Yamaha MLA8; Yamaha Music France, Marne-la-Vallée, France) linked to the video capture card (Osprey-450e) of a PC that synchronized audio and video signals. Behaviour was recorded using a camera (BUL520; brand, Active Media Concept, Vallauris, France) positioned in front of the set-up.

To avoid any effect of novelty, a hermetic jar (20 cm × 11.7 cm × 25 cm) that hosted the intruder fish when the visual live stimulus was presented (see below) was left in the experimental tank during the five days of testing. A plastic tube (TYGON® R-3603; ID = 2.00 mm; Saint-Gobain Performance Plastics Verneret, La Mothe-Aux-Aulnaies, France) hooked up to the hydrophone was used during chemical presentation. When no chemical stimulus was sent, a piece of the same plastic tube was left during the week of experimentation. The presentation of chemical stimuli was either controlled through a peristaltic pump (IPC high precision tubing pump with planetary drive; ISMATEC, Switzerland) or an aquarium pump (Tecatlantis EasyFlux300 and EasyFlux600; Aquatlantis, Guimarães, Portugal) adjusted with a flow rate of about 1 L/10 min (*Kobayashi, 2002*). Both pumps were located outside the sound-proofed chamber (see Fig. 1 for a complete view of the set-up).

## Social hierarchies' assessment in community tanks

Only males with a dominance index above 0.7 (range from 0-subordinates, to 1-dominants) were selected for experiments or chemical sampling. To establish social hierarchies between
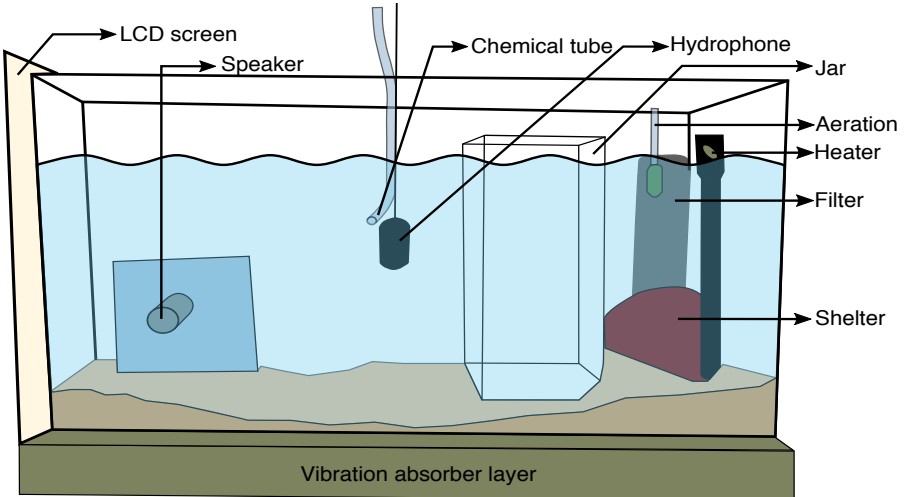

**Figure 1** **General experimental design.** Dominant males were placed in the experimental aquarium forty-eight hours before the beginning of the experiment. First, the visual contact with the social group was interrupted through the LCD screen. The screen was switched on and was behind a removable opaque partition (no image was presented). Trials started with the control period (without stimulus) during five minutes, followed by the stimuli presentation during ten minutes: (1) Teleost Ringer solution, (2) urine from dominant males, (3) real opponent in the jar, (4) agonistic video playback, (5) agonistic sound playback or (6) holding water from dominant males. In experiment 1, fish received treatment (1)–(5) (one treatment per day) on consecutive days in a random order. In experiment 2, fish were exposed to treatment (6) only.

males in stock tanks, we observed the group during ten consecutive minutes before each sampling or experiment. Following *Barata et al. (2007)*, a dominance index (DI) was used to quantify males' social rank. DI was calculated using the ratio between the number of dominant behaviours (aggressive displays or nest-digging) and the number of dominant behaviours plus the number of submissive behaviours (escape from an aggressive opponent).

## Protocol for stimuli presentation

Twenty-two dominant males (standard lengths (SL) ranging from 6.5 to 10.9 cm; mean $= 9.3$ cm) were used to test behavioural responses to each sensory channel. They were placed in the experimental aquarium 48 h before the beginning of stimuli presentation. This study consisted in six different treatments: Teleost Ringer solution (1), urine from dominant males (2), real opponent in the jar (3), agonistic video playback (4), agonistic sound playback (5) and holding water from dominant males (6). In experiment 1, fourteen fish received treatment 1–5 (one treatment per day) on consecutive days in a random order. In experiment 2, nine fish were exposed to treatment 6 only. Experiment 2 was designed to develop another chemical stimulus (holding water) to achieve the best bioassay (holding water or urine) to test this channel. Holding water was tested in a separate experiment from urine to avoid any confounding effects from any potential chemical cues that lingered in the test tank. Different males were used in the two experiments except for one male that was used in both. After experiments, all subject males returned to their original tank and

the whole set up was rinsed with water mixed with white vinegar to eliminate any potential chemical cues in the aquarium, before new males were placed in the experimental tanks.

Each experiment lasted 15 min. Five minutes before the beginning of the test, filter, aeration, and internal heater were switched off. The visual contact with the social group was interrupted through the Samsung's 172x LCD 17'' monitor used during video playback. The screen was switched on but covered by a removable opaque partition. At the same time, the loudspeaker was switched on and the real plastic tube for chemical presentation replaced the lure when necessary. The experiment started with a five minute control period while no stimulus was presented. This was followed by a ten minute period of stimulus presentation, in which either sound, chemical, or visual stimulation occurred. Details of the treatments were as follows:

**Treatment 1.** Teleost Ringer solution: 175 µL of Ringer solution (Fresh water Teleost Ringer composition was obtained from the Biological Bulletin Compendia website) was presented four times during the stimulus presentation, when the subject fish approached or touched the pump's tube.

**Treatment 2.** Urine from dominant males: After assessing the fish's social rank, urine was collected by gently squeezing the abdomen from the anterior area to the genital papilla, following the method described in *Barata et al. (2007)*. Once sampled, the fish was immediately placed back in its community tank. The urine from each male was stored at −80 °C until use for the experiments. The collected amounts of urine from dominant males (DI > 0.7) ranged from 5 µL to 340 µL with a mean of 97 µL. The chemical composition of the urine was not analysed.

During the stimulus presentation, 175 µL of diluted urine (1:2 dilution with teleost Ringer solution) was delivered four times when the fish approached or touched the pump's tube. Samples came from the same dominant male, but collected at least on two different dates, and were randomly assigned from six different dominant males (DI > 0.7). To avoid any familiar effect, the urine came from a male unknown to the subject male, i.e., from a different community tank.

**Treatment 3.** Real opponent in the jar: a male, intermediate or dominant (DI > 0.3), matched in size with the subject, was introduced in the jar located inside the subject's tank, for ten minutes of presentation without previous acclimatization.

**Treatment 4.** Agonistic video playback: Video playback consisted of three sequences from the same stimulus male: two different biting sequences and one swimming sequence. The two biting sequences were each composed of two bites and lasted five seconds. All sequences (biting and swimming) were recorded with a Logitech c 910 webcam (Logitech HD Pro c910, 1080p) and played back interactively (see below). They were filmed frontally through a one-way mirror associated to a LED light (LED light; wave 5W, TETRA, France) for the biting sequences and a 11 W 230 V light bulb for the swimming sequence. Both biting and swimming sequences were randomly selected from four males with SL ranging from 7.1 to 10.7 cm (mean = 9.4 cm). To avoid any familiar effect, fish used to create the video playback came from different tanks than the subjects. The stimulus male and subject male were matched in size (i.e., SL ratio ≤ 7%).

At the beginning of the video playback session, the opaque partition was removed and the swimming sequence was immediately played back. As soon as the subject turned to the screen, one of the two biting sequences was played back. If the individual stayed in that position, the second biting sequence was played back and so on until the two different biting sequences were played back five times each. As soon as the tested fish stopped looking at the screen, the swimming sequence was put on the screen.

**Treatment 5.** Agonistic sound playback: Agonistic sounds were played back (115 dB re. 1 μPa at 3.8 cm, within the range of the natural male sounds) with custom-made underwater loudspeakers (frequency response ± 1.5 dB in the range 20 Hz–2 kHz) and a driver developed by *Fonseca & Alves (2012)*. Sound stimuli were fed to the driver by a D/A converter (Edirol UA-25; Roland, Shizuoka-ken, Japan) controlled by Adobe Audition 3.0 (Adobe Systems Inc., Mountain View, CA, USA) on a laptop.

Two agonistic sounds of the same male were used as stimuli per experiment. These sounds were randomly assigned from six different males with SL ranging 6.0 cm–10.0 cm (mean $= 9.4$ cm) from the ENES sound archive (see *Bertucci et al., 2012*) and from new recordings made at the beginning of the study. All agonistic sounds presented 7–9 pulses (mean $= 8.0$ pulses) and a pulse rate of $18 \pm 7$ pulses per second. The pulses mean dominant frequency of the sounds was 289 Hz $\pm$ 133 Hz (see *Bertucci et al. (2012)*, for the description of sounds analysis). Pulses dominant frequencies were obtained with the 'dfreq' function from the 'seewave' R package (*Sueur, Aubin & Simonis, 2008*). As in treatment 4, playback was made interactively, simulating close interactions with a potential opponent. The stimulus was played back only when the tested individual approached the loudspeaker area, i.e., within 1–5 cm in front of the loudspeaker. Each sound was played back five times in a random order, which represented ten agonistic sounds in a test.

**Treatment 6.** Holding water from dominant males: After quantifying the social rank, dominant males (Di > 0.7) were selected and placed individually in a jar containing two litres of aerated water at 25 $\pm$ 1 °C during three hours. One litre of this solution was used during chemical presentation in the same day, to avoid any chemical degradation. We can assume the presence of urine in the holding water. Indeed, for the purposes of another ongoing study on chemical communication in the same fish, we injected three males with patent blue V (BDH Prolabo Chemicals, Fontenay-sous-Bois, France). Those males with an unknown status were small, median and large fish. Thirty minutes after the injection, we observed that fish were urinating blue and three hours after, we noted a complete blue coloration of the 2 L-aquarium's water. This demonstrates that the fish urinated during those three hours.

For this treatment, one litre of holding water was delivered continuously during the ten minutes of stimulus presentation. Holding water was obtained from eight different dominant males (Di > 0.7) and the water from one stimulus male was randomly assigned to subject males. As for urine, the holding water used came from a male unknown to the subject male.

## Behavioural analysis

Videos were analysed using JWatcher software (v. 1.0). The aggressive behaviours (i.e., lateral displays, quivers, bites, darts and sounds produced) were counted and then
summed to produce an aggressive score (*Miguel Simões et al., 2008*). We quantified the number of up-and-down swimming (when a territorial fish was swimming up and down against the wall of the aquarium). We also quantified the tank's exploration score, for which the aquarium was artificially divided in three equal zones and the number of shifts between the three zones was counted. Finally, the nest maintenance behaviour (when a territorial fish was moving sand within its territory) and the number of the shelter's visit were also counted.

## Statistics

All statistical tests were performed using R software (*R Core Team, 2016*). For each experiment, we analysed the influence of the treatments on five selected behaviours (agonistic, up-and-down swimming, tank's exploration, maintenance and the shelter's visit) separately.

Generalized linear mixed models ('glmer' functions from the 'lme4' R package, *Bates et al., 2014*) were used to assess the effect of the different treatments on the five selected behaviours. For models with Poisson family, overdispersion was tested, and if the model presented overdispersion we used a negative binomial model ('glmmADMB' R package, *Skaug et al., 2013*).

The following model was applied in experiment 1 for the five selected behaviours: **Model 1** <- Behaviour score ~ treatment * period + offset (log (Recording duration)) + (1| day) + (1| subject).

And the following model was applied in experiment 2 for the five selected behaviours: **Model 2** <- Behaviour score ~ period + offset (log (Recording duration)) + (1| subject).

**Model 1** tested the two interacting factors (treatment and period). The variable treatment had five levels. For both models, the variable period had two levels: the control and the stimulus presentation. Two random factors were added to the models: the day of the experimentation (1| day) for **Model 1** and the subject identity (1| subject) for **Model 1** and **2**. An 'offset' took into account the different recording durations between the periods of control and stimulus presentation. *P-values* were assessed using the 'Anova' function ('car' R package, *Fox & Weisberg, 2011*). In **Model 1**, if the effect of the variable treatment alone or the effect of the interaction between factors was significant, post hoc tests using Tukey's adjustment were performed using 'lsmeans' function ('lsmeans' R package, *Lenth, 2016*).

## Ethical note

All procedures described in this manuscript were conducted in accordance with appropriate French national guidelines, permits and regulations regarding animal care and experimental use (Approval no C42-218-0901, ENES lab agreement, Direction Départementale de la Protection des Populations, Préfecture du Rhône).

## RESULTS

### Experiment 1 — effect of chemical, visual and acoustic stimuli

The interaction between the treatment and the trial period had an impact on agonistic behaviour (Fig. 2A). The view of a real opponent increased the number of agonistic

behaviours observed during the stimulus presentation compared to the control period where no visual stimulus was present (Real opponent: $z = 4.252$, $P < 0.001$). Moreover, the number of agonistic behaviours during the presentation of a real opponent was significantly higher than the number of agonistic behaviours observed during the presentation of all the other treatments (Teleost Ringer solution: $z = 3.378$, $P = 0.005$; Urine from dominant males: $z = 3.300$, $P = 0.005$; Agonistic video: $z = 4.628$, $P < 0.001$; and Agonistic sound: $z = 3.334$, $P = 0.005$).

The interaction between the treatment and the trial period also had an impact on the up and down swimming (Fig. 2B). The view of a real opponent significantly decreased the number of up and down swimming during the stimulus presentation in comparison with the presentation of three other treatments: teleost Ringer solution ($z = -2.750$, $P = 0.051$), dominant males' urine ($z = -3.074$, $P = 0.051$) and agonistic sound ($z = -2.745$, $P = 0.051$).

When a real opponent was present, individuals had the tendency to increase tank's exploration during the stimulus presentation in comparison with the control period where no opponent could be seen (Fig. 2C, $z = 3.058$, $P = 0.056$).

Finally, we did not find any evidence that the number of shelter's visits (Fig. 2D, $\chi2 = 2.055$, $P = 0.726$) and maintenance behaviour (Fig. 2E, $\chi2 = 8.292$, $P = 0.081$) were influenced by treatment or trial period.

### Experiment 2 — effect of holding water

Holding water had no significant effect on agonistic behaviour (Fig. 3A, $\chi2 = 0.542$, $P = 0.462$). However, it significantly increased the number of up and down swimming (Fig. 3B, $\chi2 = 9.161$, $P = 0.002$), tended to increase the tank's exploration (Fig. 3C, $\chi2 = 2.922$, $P = 0.087$), but decreased significantly the number of visits to the shelter (Fig. 3D, $\chi2 = 8.605$, $P = 0.003$). We found no evidence that maintenance behaviour (Fig. 3E, $\chi2 = 0.657$, $P = 0.418$) was influenced by holding water.

## DISCUSSION

Our study aimed to test behavioural responses during agonistic interactions to unimodal chemical, visual or acoustic signals in a social fish and to optimise a bioassay to study multimodal communication. Here we showed that only the visual channel elicited changes in the behaviour of dominant *M. zebra* males, including aggression and general activity (explorative behaviour and swimming up and down).

### Unimodal signals

We found that the view of a real opponent provoked a highly aggressive reaction from the subject male. It also tended to increase the tank's exploration, which could be related to the subject attempting to reach the opponent confined in a jar. Subject males interacted with the intruder from three sides of the jar, ending up making more shifts between the aquarium zones. In addition, probably because they interacted for long periods with the intruder, the number of up and down swimming decreased. This aggressive response is in accordance with previous studies where visual cues alone were enough to elicit aggressive

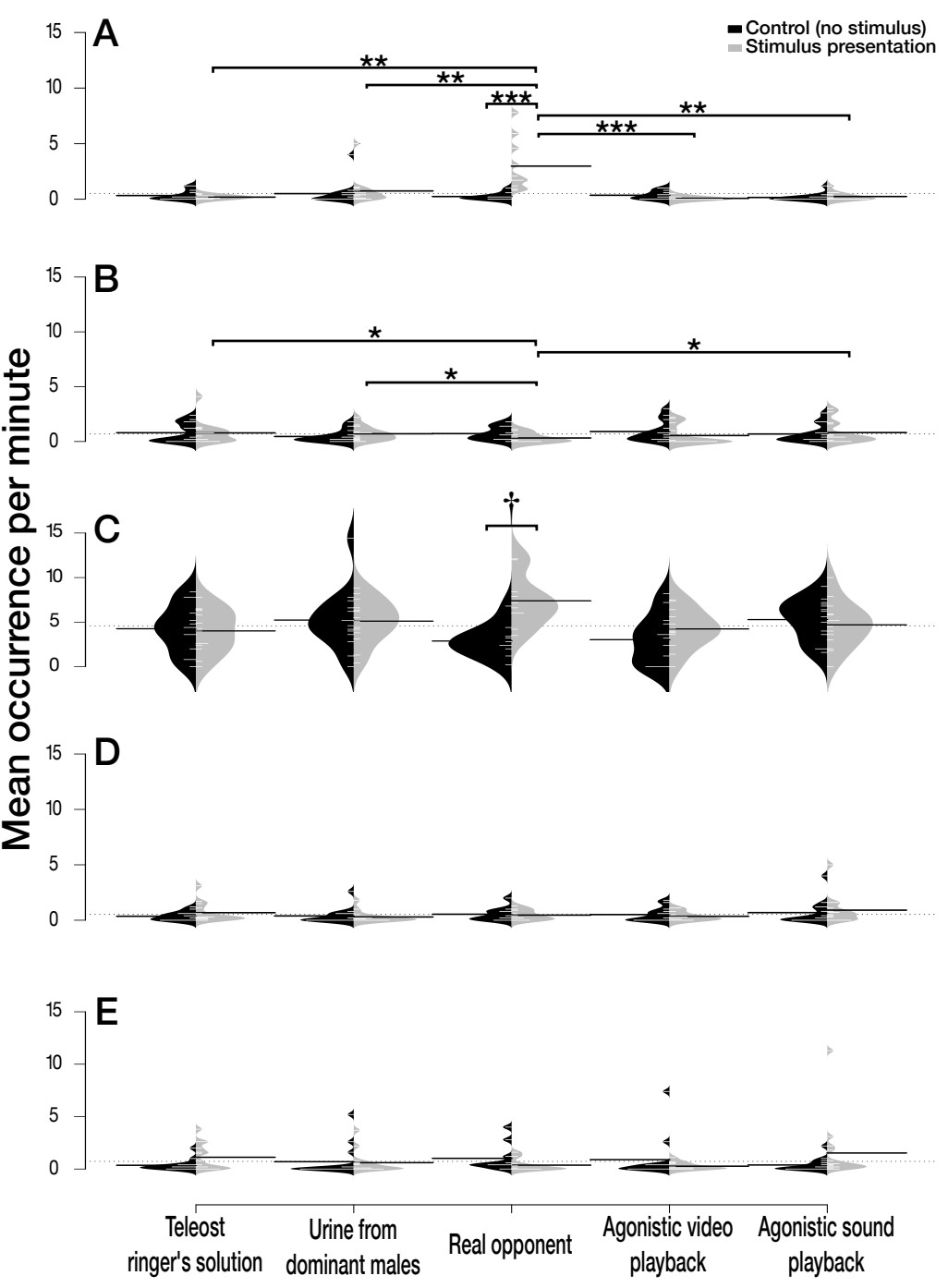

**Figure 2 Effect of the treatments on five selected behaviours.** Effect of the treatments (Teleost Ringer solution, urine from dominant males, real opponent, agonistic video playback or agonistic sound playback) on the number of (A) agonistic behaviours, (B) up and down swimming, (C) tank's exploration, (D) shelter's visit, and (E) maintenance. Each treatment included two periods, the control period without stimulus (in black) and the stimuli presentation (in light grey). Beanplots ('beanplot' R package (*Kampstra, 2008*)) combine individual observations (white lines), dataset distribution, the overall dataset average (dashed horizontal line) and the average for each subset (heavy horizontal black line). $^{\dagger}P \leq 0.1$; $^{*}P \leq 0.05$; $^{**}P \leq 0.01$, $^{***}P \leq 0.001$.

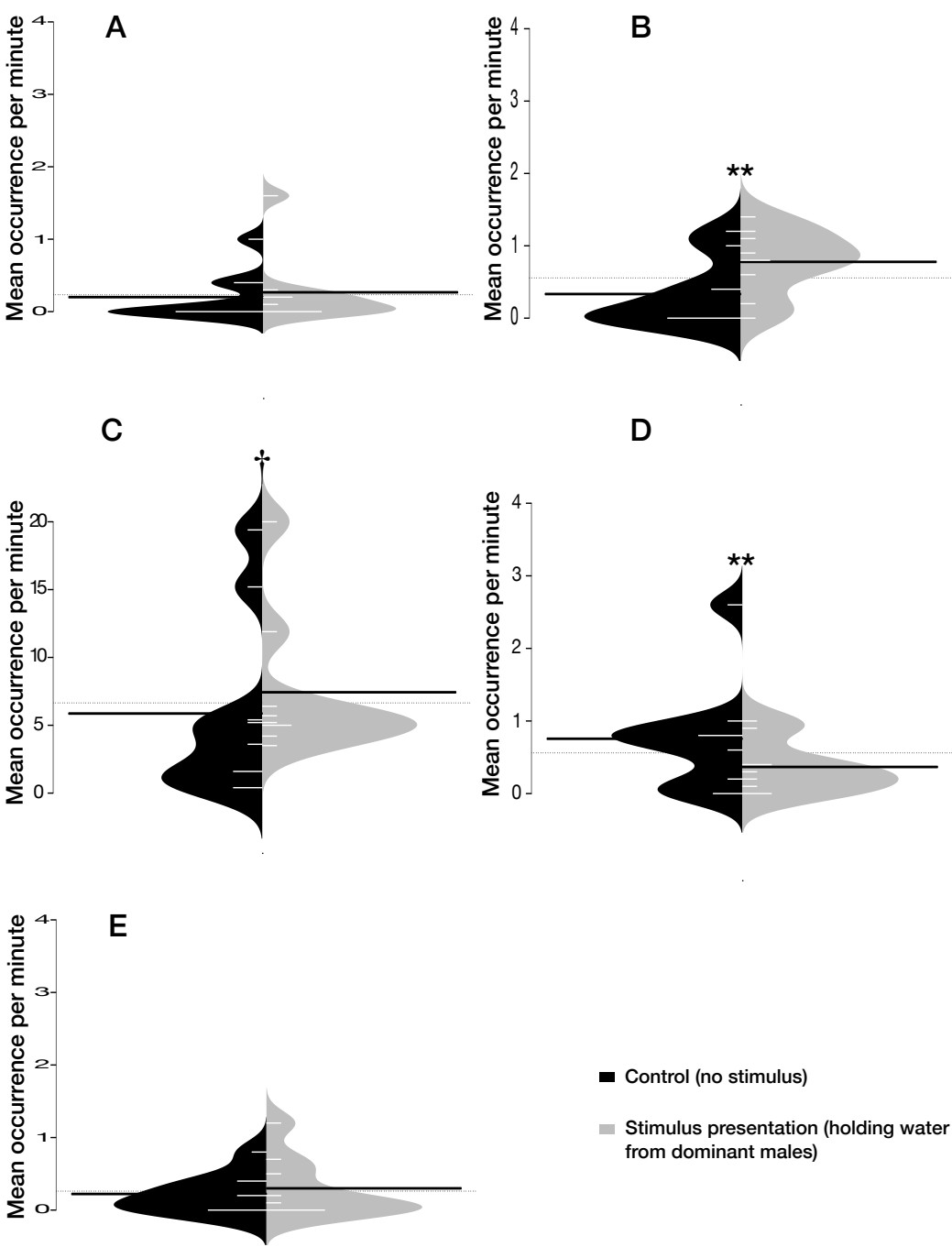

**Figure 3** **Effect of the holding water on five selected behaviours.** Effect of the holding water from dominant males on the number of (A) agonistic behaviours, (B) up and down swimming, (C) tank's exploration, (D) shelter's visit, and (E) maintenance, during the control period without stimulus (in black) and the stimulus presentation (in light grey). Beanplots ('beanplot' R package (*Kampstra, 2008*)) combine individual observations (white lines), dataset distribution, the overall dataset average (dashed horizontal line) and the average for each subset (heavy horizontal black line). $^{\dagger}P \leq 0.1$; $^{**}P \leq 0.01$.
behaviour from the subject male. For example, in *Bertucci et al. (2010)*, while an acoustic playback alone did not change the subject's behaviour in comparison to the control period in male *M. zebra*, the only view of a live contestant provoked a high aggressive answer from the tested male.

In the present study, all behavioural responses to unimodal chemical and acoustic treatments did not differ significantly from the control. Indeed, we did not find evidence that the tested fish reacted either to urine from dominant males, or to the playback of agonistic sounds presented alone. Concerning the acoustic treatment, this result is in accordance with previous studies which also found no reaction to sound playback alone as in fish agonistic sounds are typically produced within short distance from the receiver and associated with conspicuous visual displays (*Ladich & Myrberg, 2006*). Our first assumption was that acoustic stimulus would induce indirect territorial behaviours such as increased activity and nest building. *Myrberg & Riggio (1985)* showed that sounds coming from nearest neighbour provoked less diving displays of courtship than sounds coming from other males within the colony. But in *M. zebra,* sounds are associated with behaviours such as quivers during close agonistic interactions (*Miguel Simões et al., 2008*). It seems that sounds alone made by a contestant do not suffice to incite territorial males to be involved in aggression or territorial activities. *Butler & Maruska (2015)* showed in the cichlid *Astatotilapia burtoni* that the lateral line plays an essential role for mutual assessment of opponents during agonistic encounters. This system facilitates the use of non-contact assessment. Additional vibrational information could be essential and the role of the lateral line system during agonistic interactions in *M. zebra* clearly needs further investigation. Regarding the chemical component, *Chien (1973)* has reported the effect of chemical cues in the South American angelfish (*Pterophyllum scalare*) in a reproductive context. Chemical cues emitted by males increased spawning rates in the females. By contrast, in our study, urine from dominant males did not elicit any behavioural response from the tested males. To our best knowledge this is the first time that isolated chemical stimuli are tested in an agonistic context in fish. Other studies in fish underline that, during agonistic encounters, urine (like sound) is a signal used in close range aggressive interactions and needs concurrent visual stimuli to elicit immediate behavioural responses (*Barata et al., 2007*; *Keller-Costa et al., 2016*).

Under a theoretical framework, these results allow us to hypothesise a categorisation of unimodal signals in *M. zebra* into the proposed categories made by *Partan & Marler (1999)*. Because chemical and acoustic stimuli did not elicit any response in *M. zebra* males but visual stimuli did, unimodal signals should be categorized as non-redundant, since redundant signals have equivalent effects on a receiver. When presented as a composite signal, non-redundant signals can lead to different effects such as an independence effect. When combined, non-redundant signals provoke behavioural reactions which are the same as those observed when signals are presented alone. On the other hand, if one signal overshadows or changes the effect of the other, it has a dominance or a modulation effect. If otherwise the combination of those signals elicits an entirely new response there is an emergence effect. How different sensory modalities interplay during fish communication, in particular during agonistic interactions, remains to be unravelled. In *M. zebra* acoustic

agonistic signals seem to modulate the response to the visual channel (*Bertucci et al., 2010*). In contrast, the acoustic sensory modality seems to dominate over the visual one in a small marine goby (*Amorim et al., 2013*). In *Neolamprologus pulcher* chemical signals appear to modulate the response to visual signals during an agonistic encounter (*Bayani, Taborsky & Frommen, 2017*). Future work needs to address how the different sensory channels interact to produce an adaptive response.

## Which bioassay?

In addition to investigate the behavioural response to unimodal stimuli, we aimed to develop a bioassay to assess the role of each sensory channel in eliciting behavioural decisions. Different types of stimuli can be chosen to test the role of one signal modality with each having different pros and cons. For example, in the present study a live opponent was the treatment that elicited the strongest response from subject males compared to a video playback. However, because within one-modality a signal is usually multicomponent (*Candolin, 2003*), it is important to be able to control the other parameters, apart from the ones that are being tested. Although we controlled for size effects (live intruders were size-matched to the tested fish) we could not control other aspects such as changes in body colour, behaviour, etc. Because of these drawbacks, other studies have used different visual stimuli such as mirror tests (e.g., *Raffinger & Ladich, 2009*) or visual computer-manipulated stimuli (*Watanabe, Shinozuka & Kikusui, 2016*). Although more and more research uses computer animation as a tool to control visual cues, including in cichlids (*Balzarini et al., 2016*), our results underlined an absence of response towards video playback. This was recently the case in another study using cichlids (*Wackermannova et al., 2017*). The use of RGB video screen, the screen frame rate (developed for human sight), the lack of UV components and three-dimensional depth are constraints that may affect visual perception and could explain this absence of response. In any case, natural motion patterns, velocity and interaction should not have been affected here as video playbacks consisted of recordings from real individuals either biting the tank's glass with a one-way mirror, or performing routine swimming.

Finally, we tested another chemical cue, holding water from dominant males, which is classically used in cichlids (*De Caprona, 1974*; *Thünken et al., 2009*). Here, holding water increased the up and down swimming, tended to increase the tank's exploration and decreased the number of shelter's visits. Unlike urine, holding water seems to make the fish aware of the potential presence of a contestant even though no visual signal was present. Holding water represents a more diffuse cue than urine, as urine is generally used during close range aggressive interactions (*Barata et al., 2007*; *Escobar-Camacho & Carleton, 2015*). It also contains in addition to urine, other components coming from the gills, the mucus, and the faeces. These differences could explain that urine without visual signal had no effect on *M. zebra* males' behaviours while holding water without visual signal had an impact. Consequently, the best bioassay in testing behavioural answers in *M. zebra* to chemical signals seems to be the use of holding water. It is a lot easier and less intrusive to collect than pure urine, where collected amounts varied from nothing to very small quantities (<340 µL). Moreover, the handling provokes a stressful response from the handled fish,

typically accompanied by increased circulating cortisol (*Ramsay et al., 2009*). Because we left the fish in a two litres tank during three hours to obtain holding water, the manipulation is less stressful and we can manage larger liquid quantities. Nevertheless, if the question is to seek the existence of active molecules in urine during agonistic encounters, then sampling pure urine should be more promising.

In conclusion, *M. zebra* territorial males seem to rely mostly on the visual sensory modality, as the view of a real opponent provoked intense aggressive reaction but the acoustic signal and chemical cue alone did not elicit obvious changes in their immediate behaviours. The different sensory channels seem to be non-redundant (*sensu Partan & Marler, 1999*) in the study species and likely in most other fishes (*Ladich, 2004*), but how different modalities are integrated to produce adaptive behaviour in fish remains to be answered. Our results also emphasize the need to optimise experimental protocols to test the significance of multimodal communication during specific behavioural contexts, to better understand the evolution of signalling across vertebrates (*Partan & Marler, 1999*; *Candolin, 2003*; *Narins et al., 2005*).

## ACKNOWLEDGEMENTS

We thank Friedrich Ladich and an anonymous referee for their valuable comments.

### Funding
This study has been funded by the Université de Lyon/Saint-Etienne and the Centre National de la Recherche Scientifique (CNRS). LC was supported by a PhD fellowship from the French 'Ministère de la Recherche'. Travel between France and Portugal was funded by the Université de Lyon/Saint-Etienne. The funders had no role in study design, data collection and analysis, decision to publish, or preparation of the manuscript.

### Grant Disclosures
The following grant information was disclosed by the authors:
Université de Lyon/Saint-Etienne.
Centre National de la Recherche Scientifique (CNRS).
French 'Ministère de la Recherche.

### Competing Interests
The authors declare there are no competing interests.

### Author Contributions
- Laura Chabrolles conceived and designed the experiments, performed the experiments, analyzed the data, wrote the paper, prepared figures and/or tables, reviewed drafts of the paper.
- Imen Ben Ammar, Joël Attia, M. Clara P. Amorim and Marilyn Beauchaud conceived and designed the experiments, reviewed drafts of the paper.

- Marie S.A. Fernandez analyzed the data, reviewed drafts of the paper.
- Nicolas Boyer conceived and designed the experiments.
- Paulo J. Fonseca conceived and designed the experiments, contributed reagents/materials/analysis tools.

## Animal Ethics

The following information was supplied relating to ethical approvals (i.e., approving body and any reference numbers):

All procedures described in this manuscript were conducted in accordance with appropriate French national guidelines, permits and regulations regarding animal care and experimental use (Approval no C42-218-0901, ENES lab agreement, Direction Départementale de la Protection des Populations, Préfecture du Rhône).

## Data Availability

Raw data is available in the Supplemental Information.

## Supplemental Information

Supplemental information for this article can be found online at http://dx.doi.org/10.7717/peerj.3643#supplemental-information.

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
