# Peer review of "Appraisal of unimodal cues during agonistic interactions in Maylandia zebra"

_PeerJ, doi:10.7717/peerj.3643_

## Round 0.1 · original submission · Major Revisions

The reviewers prepared several comments and queries for your attention, and I have attached a copy of their reports below. I believe that all comments will improve your manuscript quality. I suggest responding to all queries point-by-point.

·

Basic reporting

No comment

Experimental design

No comment

Validity of the findings

No comment

Additional comments

Multimodal signaling is most likely important during reproductive activities and conflict resolution in the majority of animals in general and fishes in particular. The present study tries to present evidence that this is also the case in the zebra mbuna Metriaclima zebra an African representative of the cichlid family.

This is an important topic in fishes in general and therefore I very much suggest that authors do not jump from the general introduction within in first two paragraphs to the African species of a single fish family but instead mention that these communication channels are known and most likely used in many more of the currently known 500+ fish families.

Similarly, they should mention that even among African cichlids there are more signal types used besides the visual, acoustic and chemical ones. Butler and Maruska (2015) showed that vibrational stimuli detected by the lateral line are used by African cichlid Astatotilapia burtoni for mutual assessment of opponents. This signal type should be added to the introduction and its potential importance for fishes in general and cichlids in particular should be discussed.

Butler JM, Maruska KP. 2015. The mechanosensory lateral line is used to assess opponents and mediate aggressive behaviors during territorial interactions in an African cichlid fish. Journal of Experimental Biology 218:3284-3294.

Specific comments

Methods

Lines 142-143: I suggest that the word “playback” should not be used when presenting non-acoustic stimuli because this is confusing. Playback is clearly an acoustic term and chemical playbacks, visual playbacks or acoustic playback sound awkward. Why not use the terms visual, acoustic and chemical stimuli presentation, etc.?

Line 168: Please add how much urine was typically sampled from a dominant male. Explain if the chemical composition of the urine was analyzed. Do authors have any idea if and which pheromones may be involved in chemical signaling.

Line 205: Treatment 5:
Describe stimuli for sound playbacks in more detail. Give the sound pressure levels (SPL) in dB re 1 µPa and the distance (range) between the speaker and the tested individual fish.

Line 214: Treatment 6:
Explain if the holding water from dominant males contains any urine or if it is assumed that males do not urinate outside agonistic interactions.

Results

Line 274: I do not understand this sentence. Does the term “playback period” mean that a real behaving animal is presented to the fish investigated? Why not describe it more clearly? Playback typically means that a sound has been recorded before and played back afterwards.

Discussion

Line 327: It should be mentioned that playbacks of sounds can elicit territorial behaviour in fish. Myrberg & Riggio (1985) showed that playbacks of nonresident’s sounds elicited dipping behaviour in a territorial fish.

Myrberg AA, Riggio RJ. 1985. Acoustically mediated individual recognition by a coral reef fish (Pomacentrus partitus). Animal Behaviour 33:411-416.

References:

I suggest writing all species names in italics in addition to that in line 413.

Line 477: Correct citation; the publisher and city are: Narosa Publishing House, New Delhi.

Line 479: Correct citation: “Communication in Fishes” is not a journal but a book. Thus add editor, publisher, place of publisher, etc

Reviewer 2 ·

Basic reporting

No comment

Experimental design

1. How authors controlled how many times each fish was exposed to the stimulus chemical, sound and screen during "the ten minutes’ playback period". All fishes were exposed for at least one moment?

2. “Real opponent in the jar: a male, regardless his status, matched in size with the subject, was introduced in the jar located inside the subject’s tank…” How authors controlled if the fishes were exposed to dominant males or submissive males? An equal number of tested fishes were exposed to dominant males and submissive males?

Validity of the findings

1. Considering that acquiring mates is an important cause of agonistic behavior, the absence of female fish during stimulus and test could not been influence obtained data? The authors should clarify this issue.

2. Authors inform that “urine was delivered four times when the tested fish approached or touched the pump’s tube”and “agonistic sounds were played back only when the tested individual approached the loudspeaker area”. Also, “screen biting sequences was played back when tested fish turned to the screen”. The occurrence of these stimuli depends of an initial exploratory behavior of the tested fishes. Differently from chemical, sound and video stimulus, real opponent presence occurred constantly for ten minutes. Could the fact that real opponent presence has been the unique constant stimulus influenced the obtained result? Authors should explain this question.

Additional comments

No additional comments

---

## Round 0.2 · accepted · Accept

I am pleased to inform you that your manuscript referenced above has been accepted for publication in PeerJ.

·

Basic reporting

O.k.

Experimental design

O.k.

Validity of the findings

O.k.

Additional comments

All corrections have been carried out satisfactorily.

Reviewer 2 ·

Basic reporting

No comment

Experimental design

No comment

Validity of the findings

No comment

Additional comments

The authors clarified all the questions that have been pointed.